# The Effects of Different Supply Chain Integration Strategies on Disruption Recovery: A System Dynamics Study on the Cheese Industry

Quan Zhu [1,*], Harold Krikke [2] and Marjolein C. J. Caniëls [2]

1. Sustainable International Business, International Business School Maastricht, Zuyd University of Applied Sciences, 6217 HB Maastricht, The Netherlands
2. Faculty of Management, Open Universiteit, 6419 AT Heerlen, The Netherlands; Harold.Krikke@ou.nl (H.K.); Marjolein.Caniels@ou.nl (M.C.J.C.)
* Correspondence: quan.zhu@zuyd.nl

**Abstract:** Long and complex supply chains are vulnerable to disruptions. One way to solve this problem is to successfully manage supply chain integration (SCI). A system dynamics simulation is thus applied to study a cheese supply chain with three individual firms: a producer, a logistics service provider (LSP), and a retailer. Our purpose is to study the effects of SCI strategies with different dimensional focuses, i.e., information integration (Scenario 1), relational integration (Scenario 2), and operational integration (Scenario 3), on the recovery of three types of disruptions, i.e., a producer capacity disruption, an LSP capacity disruption, and a demand disruption. Tests of parameter scenarios are further applied to provide solutions for supply chains using different strategies. Our results indicate that Scenario 3 is the best practice, regardless of any type of disruption, while Scenario 1 usually achieves the worst performance. This is consistent with an evolutionary perspective of supply chain integration: information integration gives firms a competitive advantage as the first step. Working as partners to share the most appropriate information leads to greater benefits. We extend this perspective by showing that further elimination of information delay helps the supply chain achieve the best performance.

**Keywords:** supply chain integration; system dynamics; disruption recovery; cheese industry





## 1. Introduction

Globalization and rapid development of information technology are changing today's inter-organizational relationships [1]. Firms increasingly depend on a complicated network of global partners to deliver products in the right quantity, at the right place and time, and under persistent cost pressures [2]. Unfortunately, long and complex supply chains are usually slow in responding to changes and, hence, they are vulnerable to supply chain disruptions [3–5]. Supply chain disruptions can transmit among supply chain members [6] and their magnitude can be significantly influenced by supply chain rippling and network effects [7], hence they should be understood and managed as a whole from end to end [8]. One way to achieve this goal is thus to successfully manage supply chain integration (SCI), which requires cross-firm business processes with appropriate levels of information sharing, trust and commitment, as well as operational coordination [9].

Many studies suggest that SCI has a positive impact on performance [10,11]. However, in a literature review [12], 12 out of 31 of the reviewed papers show that more SCI does not always improve performance. One reason may be that different dimensions of SCI were considered. For example, information and knowledge sharing may be not consistently related to effective coordination of manufacturing activities across firms [13]. Relational mechanisms (e.g., trust and commitment) are needed to reinforce partner co-operation and mitigate risks arising from unanticipated events [14,15]. An evolutionary

perspective of SCI shows that information integration may have given firms a competitive advantage in the early- to mid-1990s. It constituted a first step in SCI, but may not be sufficient to excel. Working as partners, rather than simply transferring information, leads to the greatest benefits [16].

Prior research on SCI has typically drawn on survey-based studies [17–19]. They primarily focus on a general or mixed view of SCI rather than looking at SCI strategies with different dimensional focuses varying from information technology to relationships to operations. Furthermore, current survey-based studies are static while supply chain disruptions always deal with dynamic events that develop over time [20]. This drawback is shared with traditional optimization models [21,22], which calculate the static equilibrium to search for an optimal solution for an operational problem [2]. It is highly advisable to use a system dynamics simulation to study the dynamic SCI strategies [23].

The contribution of this paper is to investigate the effects of SCI strategies with different dimensional focuses on disruption recovery. Our purpose is to understand the dynamics of different SCI strategies, test the evolutionary perspective of SCI, and provide suggestions for supply chains using different SCI strategies in the context of disruption recovery.

This paper is structured in the following way. In Section 2, we review the literature of supply chain disruptions, dimensions of SCI, and system dynamics. Section 3 illustrates simulation models for a cheese supply chain. The results of the simulation are discussed in Section 4. Finally, the paper concludes with discussion, implications, and suggestions for future research in Section 5.

## 2. Literature Review

### 2.1. Supply Chain Disruptions

In the last few years, supply chains have become more vulnerable to disruptions [24–27]. Accidents become inevitable or even normal in complex and tightly coupled systems, as they extend the focal firm's context to its supply chain's context [28]. It is not surprising that, with high clock speed, long and complex supply chains have become prone to disruptions [29].

Supply chain disruptions are defined in the literature as unplanned and unanticipated events that disrupt the normal flow of goods and materials within a supply chain [26,30,31]. Based on this definition, Wagner and Bode [26] proposed five different sources of disruptions: demand side, supply side, legal/regulatory, infrastructure, and catastrophes. Although disruptions may have different sources and may not occur independently, all the disruptions will in the end lead to one or several problems on product supply, distribution, and demand.

### 2.2. Dimensions of Supply Chain Integration (SCI)

The review of dimensions of SCI is important, as different dimensional perspectives generate diverse recovery strategies. According to Alfalla-Luque et al. [1] and Leuschner et al. [9], three independent dimensions of SCI are identified: information integration, relational integration, and operational integration. When firms engage in SCI, they first share data and information [32–34]. Information integration refers to the coordination of information transfer, collaborative communication, and supporting technology among firms in the supply chain [9]. The flow of information directly impacts production plans, inventory control, and distribution plans [35]. Through an agent-based simulation, Breuer et al. [36] have created a model that reflects the role of information integration on maintaining the flow of goods after disruptions. Via a simulation-based simulation, Di Nardo et al. [37] have developed a stochastic dynamic optimization model to guide the information system integration to optimize material flow in a full-service approach, and further verified the model in a case study. However, the quality of integration heavily depends on the appropriateness of information for exchange, which is based on trust and inter-dependence among supply chain partners [38]. Thus, different from infor-

mation integration, relational integration refers to the adoption of a strategic connection between firms in the supply chain that is characterized by trust and commitment [9]. Through relational integration, supply chain partners share the most proper information. Nagurneya and Qiang [39] have applied numerical examples to demonstrate how relational integration helped supply chain partners create synergy to deal with disruptions. The last dimension, operational integration, is generated when firms integrate operational activities [40]. It refers to joint activity development, collaborative work processes and coordinated decision-making among firms in the supply chain [9]. Re-organization of outsourcing, packaging customization, agreements on delivery frequency, and common use of logistical equipment have a high impact on cost, quality, and speed [1]. Through a survey of 203 manufacturing companies in Australia, Chen et al. [41] have certified that operational integration can help alleviate the impacts of supply chain disruptions.

### 2.3. System Dynamics Modeling on SCI and Disruption Recovery

System dynamics is a methodology that is capable of studying and modeling complex systems [42]. A "system" refers to a group of inter-dependent or autonomous components/entities working together for a common cause [23]. The complexity of supply chains, especially those which encompass several partners, warrants a perspective that considers the supply chain structure and the feedback inherent in these structures, which is provided by system dynamics modeling [43,44]. Although discrete event simulation can deal successfully with disruption events, system dynamics have the capability to reproduce the interaction of different system agents to improve the understanding of the real system [45], and at the same time overcome the limitation of static assessments [46]. Therefore, we select system dynamics for this research to gain deep understanding of the supply chain structure with a high level of complexity and to foresee the impacts of changes in policies (i.e., different SCI strategies in this research) on disruption recovery [47].

System dynamics modeling possesses a high frequency of use in the research of SCI [48]. Some of the research also concerned disruptions, which we discuss here. Helo [49] used a system dynamics simulation to demonstrate how agility is built into supply chains. The analysis recommends smaller order sizes, information integration, and capacity analysis as methods of improving the responsiveness after disruptions. Cooke and Rohleder [50] built a model of a safety and incident learning system to help managers move safety performance from normal accidents to high reliability. Although taking a single organization perspective, they identified that the most important feedback loops are through management/personal commitment to safety and willingness to investigate/report incidents, which reflect the idea of relational integration. Wilson [43] compared a traditional supply chain with a vendor managed inventory (VMI) system, when a transportation disruption occurs between two echelons in a five-echelon supply chain. The impacts are less severe for the VMI structure, which is a form of operational integration.

All the above-mentioned studies only discussed one dimension of SCI on disruption recovery. Therefore, this paper will measure and compare the effects of SCI strategies with different dimensional focuses on disruption recovery. In addition, we will also provide suggestions for supply chains using different SCI strategies. The former is achieved by adding/deleting some feedback loops, adding/deleting parameters, and/or changing the structure of the feedback loops (i.e., structure scenarios), while the latter is realized by adjusting parameter values (i.e., parameter scenarios) [51].

## 3. The Simulation Model and Analysis Methodology

### 3.1. Research Background

Our simulation model is derived from a business simulation game (www.bedrijfssimulaties.nl, accessed on 16 March 2021) that was developed by a Dutch university. The game was used by the second author for academic teaching. It uses data that were collected in 2011 from the real-life practice of a large multinational dairy firm. Cost structures have not been changed significantly since then. The game's learning objectives are strongly related to SCI.

Through playing the game, one can explore the interactions between supply chain goals (i.e., supply efficiency and order fulfillment) and integration efforts (i.e., three dimensions of SCI). Our simulation model resembles the gaming model but emphasizes the role of SCI on disruption recovery. Therefore, our research used original parameter values from the game as well as adjusted parameter values that are related to disruptions. The software that was used to build our simulation model is STELLA®.

The focus of this paper is on the cheese industry. The main inbound resource is milk, of which supply is usually abundant. Production start-up is pull, i.e., demand driven. Once started, the production process becomes push (i.e., supply driven) until the cheese is ready. After milk is pasteurized, it is curdled and takes its typical molded shape. Next, it is salted, which takes one to five days. The entire process takes around a week. However, most of the production lead time involves waiting. Depending on the type of cheese, the product has to mature for two to sixteen weeks. Despite its long production time, cheese is perishable. If it is not delivered to customers in time, it does not have value any more.

### 3.2. Simulation Assumptions, Types of Disruptions, and Scenarios

The cheese supply chain in our model consists of three individual firms: a producer, a logistics service provider (LSP), and a retailer. Milk and other resources are assumed to be infinite for the producer. The producer produces the cheese with a six weeks' production time and the finished products are immediately delivered to the LSP. When ready, the cheese has a remaining shelf life of six weeks. The LSP keeps inventory, but it is the ordering policy pulled by the retailer and the production volume pushed by the producer that determine the actual inventory level. This is beyond the control of the LSP. The retailer sells the products to consumers. The sales time is one week.

Table 1 shows the initial simulation inputs. For "Real customer demand rate", the value is a normal distribution with mean of $1.28 \times 10^6$ kg/week and standard deviation of $1.00 \times 10^5$ kg/week.

**Table 1.** Simulation inputs with value

| Simulation Input | Value |
| --- | --- |
| Producer order backlog | $7.68 \times 10^6$ kg |
| Production time | 6 weeks |
| Alpha, Beta | 0.5 |
| Producer base capacity, Logistics service provider (LSP) base capacity | $1.28 \times 10^6$ kg/week |
| Producer surge capacity, LSP surge capacity | $3.20 \times 10^5$ kg/week |
| Order backlog, LSP inventory, Cumulative demand | 1 kg |
| LSP target shipment time, Retailer sales time | 1 week |
| Retailer inventory | $2.56 \times 10^6$ kg |
| Producer perception delay, Communicated lead time delay, Time to adjust backlog | 2 weeks |
| Normal delivery reliability | 0.95 |
| Real customer demand rate | Normal ($1.28 \times 10^6$, $1.00 \times 10^5$) kg/week |

There are three types of disruptions that we will simulate to capture problems in product supply, distribution, and demand: a producer capacity disruption, an LSP capacity disruption, and a demand disruption. For the first two types of disruptions, only capacity shortages are considered in our model, as the impact of transportation disruptions on supply chain performance has already been demonstrated by Wilson [43]. For the last type of disruption, we only consider the situation of customer demand that is far below the available capacity, as the situation of customer demand that is far beyond the available capacity is similar to the bullwhip effect [35], of which solutions have been well discussed [35,42,52,53].

For each type of disruption, we simulate a process in 24 weeks: in the first 9 weeks, all the variables are stable; at the end of Week 9, a disruption happened and caused a base capacity or demand rate drop of $6.40 \times 10^5$ kg/week; at the end of Week 12, the base capacity or demand rate recovers $3.20 \times 10^5$ kg/week; at the end of Week 15, the base

capacity or demand rate recovers another $3.20 \times 10^5$ kg/week—back to the starting level, until the end of simulation. The length of simulation is set to be 24 weeks, because tests show that the extension does not have any actual influence on the results.

Three structure scenarios will be simulated to evaluate the effects of information integration (Scenario 1), relational integration (Scenario 2), and operational integration (Scenario 3) on disruption recovery. Each scenario needs a different model. The models of Scenarios 1 and 2 are adapted from Özbayrak et al. [42]. The difference between Scenarios 1 and 2 is whether there is information distortion. Scenario 2 shares the most proper information ("Real customer demand rate" and "Workload") without information distortion, while Scenario 1 shares "Channel demand rate" and "LSP shipment time" and thus suffers from information distortion. Although firms in Scenario 1 can collect the most proper information by themselves, it will take longer time and thus only contribute to a part of their own decision-making (illustrated by "Alpha" and "Beta"). The model of Scenario 3 is adapted from the VMI model of Wilson [43] and it is without information distortion (same as Scenario 2, and different from Scenario 1). In a VMI partnership, transactions customarily initiated by the buyer (e.g., purchase orders) are now initiated by the vendor [54]. In our simulation, the retailer is "the buyer" and the producer serves as "the vendor". Compared to Scenario 2, Scenario 3 eliminates information delay (illustrated by "Producer perception delay" and "Communicated lead time delay") to facilitate full information transparency. The illustration of structure scenarios is shown in Table 2. More details of our model structures will be presented in Section 3.3.

**Table 2.** Illustration of structure scenarios

|  | Scenario 1 | Scenario 2 | Scenario 3 |
|---|---|---|---|
| Content | Information integration | Relational integration | Operational integration |
| Information distortion | Yes | No | No |
| Distorted information | Channel demand rate, LSP shipment time | Not applicable | Not applicable |
| The most proper information | Real customer demand rate, Workload | Real customer demand rate, Workload | Real customer demand rate, Workload |
| Information used for decision-making | Alpha (or Beta) × Distorted information + [1 − Alpha (or Beta)] × The most proper information | The most proper information | The most proper information |
| Information delay | Yes | Yes | No |
| Delays | Producer perception delay, Communicated lead time delay | Producer perception delay, Communicated lead time delay | Not applicable |

In addition, within each structure scenario, our model may be sensitive to (1) "Alpha" and "Beta" that reflect information distortion and (2) "Producer perception delay" and "Communicated lead time delay" that reflect information delay. To provide suggestions for supply chains using different SCI strategies, we apply different parameter scenarios by changing "Alpha" and "Beta" from 0 to 1 (11 parameter scenarios with an interval of 0.1) and "Producer perception delay" and "Communicated lead time delay" from 1 week to 3 weeks (11 parameter scenarios with an interval of 0.2 week). Please note that the tests of parameter scenarios for "Alpha" and "Beta" are only applicable to information integration (Scenario 1), as only Scenario 1 has to decide "Alpha" and "Beta". The tests of parameter scenarios for "Producer perception delay" and "Communicated lead time delay" are not applied to operational integration (Scenario 3), as there is no information delay in this structure scenario.

*3.3. Model Structures*

In this section, we illustrate the model for each structure scenario from the producer's production to the LSP's operation and finally to the retailer's order fulfillment. According to the dynamic resource-based view, strategic assets are modelled as stocks (in rectangles) of available tangible or intangible factors in a given time. Their dynamics depend on the value of corresponding inflows and outflows [55]. Such flows are modelled as "valves" that directly change the rate of evolution of each strategic resource [56]. The source and sink of each strategic resource are represented by a cloud symbol. The source has an arrow coming out, while the sink has an arrow going into the cloud [23]. There are two typical processes in a system dynamics model [57] (pp. 411–434). One is known as a material delay, since it captures the physical flow of material through a delay process. In a material delay, the stock level accumulates at a rate equal to the difference between the inflow and outflow rates. The other kind of delay, which represents the gradual adjustment of a perception or belief, is called an information delay. Because information, unlike material, is not conserved, a bi-flow structure (i.e., that can be both increased and decreased) is needed to capture an information delay. The stock level adjusts to the actual information input in proportion to the size of the difference in one's belief. An adjustment time/delay determines how rapidly one's belief responds to the difference.

The structure of how the producer organizes production for Scenario 1 is shown in Figure 1. It consists of two information delays (for "Perceived channel demand rate" and "Perceived real customer demand rate") and one material delay (for "Producer order backlog"). "Channel demand rate" is the information directly shared by the LSP. It is not the most proper information to share, as it equals "Customer demand rate", which is a sum of "Real customer demand rate" (the most proper information) and "Customer backlog adjustments" (for the retailer's own purpose to stabilize its inventory). We assume that the producer can only realize "Real customer demand rate" one week after its managers receive "Channel demand rate" from the LSP, because they need extra time to collect the information of "Real customer demand rate" by themselves. We set "Alpha" as 0.5, which means that one half of "Product demand rate" is from "Perceived channel demand rate" and the other half (equals 1–"Alpha") results from "Perceived real customer demand rate". This setting shows the fact that the producer does not fully trust the information shared by the LSP, because "Channel demand rate" is not the most proper information. However, as collecting the most proper information (i.e., "Real customer demand rate") takes more time, the producer has to consider "Channel demand rate" as well when it decides "Product demand rate". "Product shipment rate" is the minimum value between "Producer desired shipment rate" and "Producer base capacity". In the simulation model of a producer capacity disruption, "Producer base capacity" will be replaced by "Producer base capacity" plus "Producer surge capacity". This also applies to the other two scenarios.

The producer's production simulation model for Scenario 2 is different from that for Scenario 1 on "Product demand rate". Here, "Product demand rate" is only from "Perceived real customer demand rate", which is an information delay of "Real customer demand rate" (the most proper information) that is directly shared by the LSP. In this sense, there is no "Perceived channel demand rate" or "Alpha". "Time for producer to perceive" is replaced by "Producer perception delay" with the same expression and value in use as Scenario 1.

In the producer's production simulation model for Scenario 3, the producer directly receives "Real customer demand rate" from the retailer without any information delay, which means "Product demand rate" equals "Real customer demand rate". The other parts of the model are the same as those of Scenarios 1 and 2.

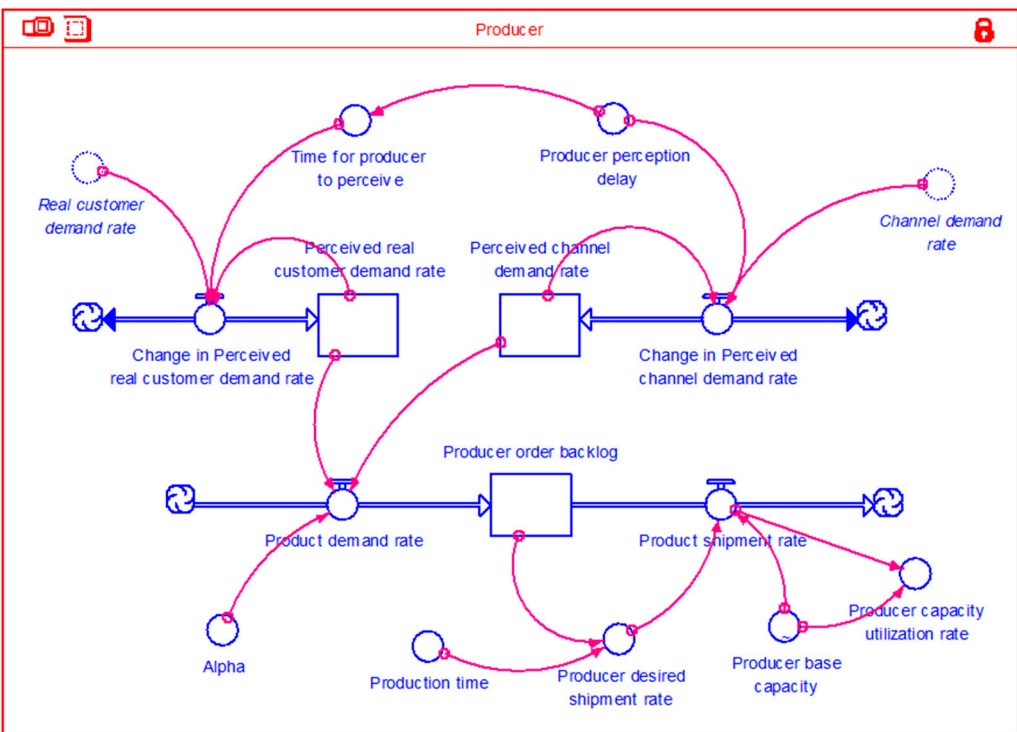

**Figure 1.** Producer's production simulation model for Scenario 1.

Figure 2 presents the structure of the LSP's operation simulation model for Scenarios 1 and 2. There are two material delays (for "Order backlog" and "LSP inventory"). "LSP shipment rate" is the minimum value of "LSP base capacity", "LSP desired shipment rate", and "LSP inventory"/"LSP target shipment time". "Workload" will be used by the retailer to judge the LSP's ability to deliver order on time in the retailer's order fulfillment simulation model. This is the most proper information to share to the retailer, as it directly shows the status of LSP capacity usage without any information distortion. "Workload" equals "LSP inventory"/"LSP target shipment time"/"LSP base capacity". "LSP shipment time" will also be used in the retailer's order fulfillment simulation model. "LSP shipment time" is equal to "LSP inventory" divided by "LSP shipment rate". Compared to "Workload", "LSP shipment time" is the less proper information to share with the retailer, as it involves information distortion that is caused by the non-linear "minimum" function in "LSP shipment rate". In the simulation model of a producer (or an LSP) capacity disruption, "LSP base capacity" will be displaced by "LSP base capacity" plus "LSP surge capacity". This also applies to Scenario 3.

In the LSP's operation simulation model for Scenario 3 (Figure 3), the LSP does not participate in the information sharing process, so there is no calculation for "Order backlog", as well as no shared information of "Workload" or "LSP shipment time" (but it is calculated as a performance measure). Here, "LSP desired shipment rate" equals "LSP inventory"/"LSP target shipment time". "LSP shipment rate" is the minimum value of "LSP desired shipment rate" and "LSP capacity". The other parts of the model are the same as those of Scenarios 1 and 2.

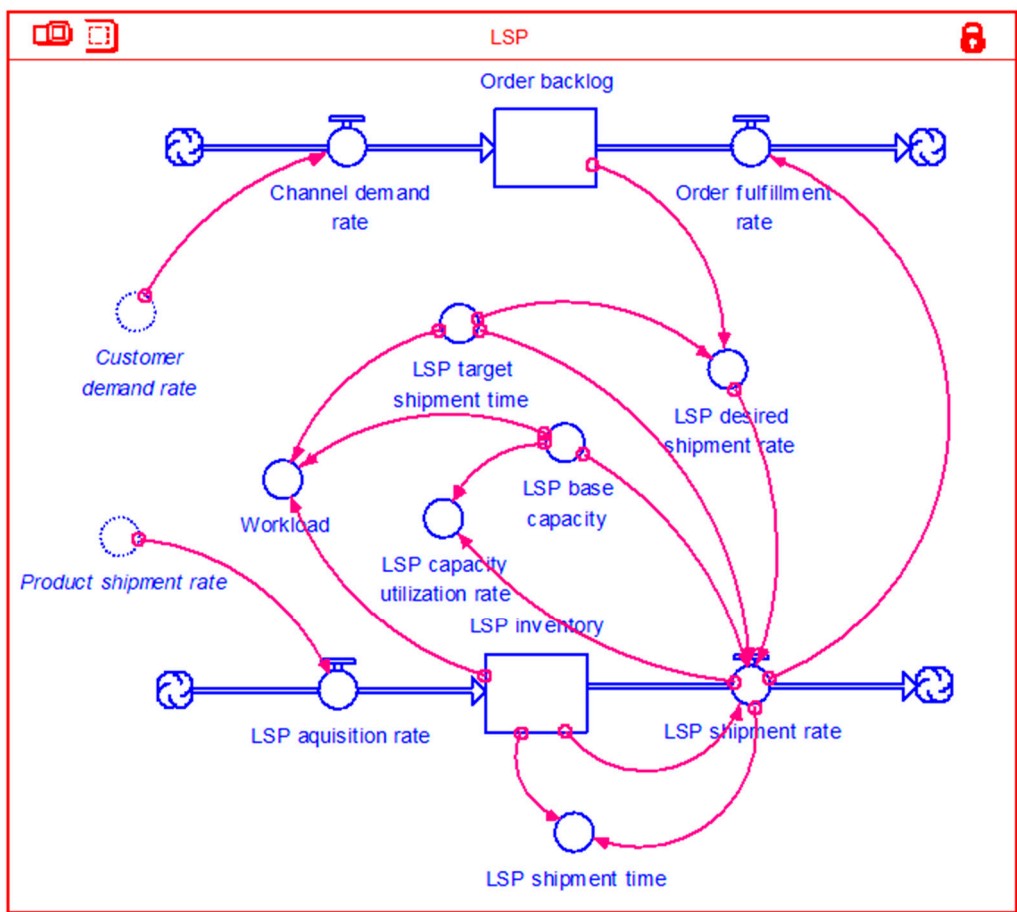

**Figure 2.** logistics service provider (LSP)'s operation simulation model for Scenarios 1 and 2.

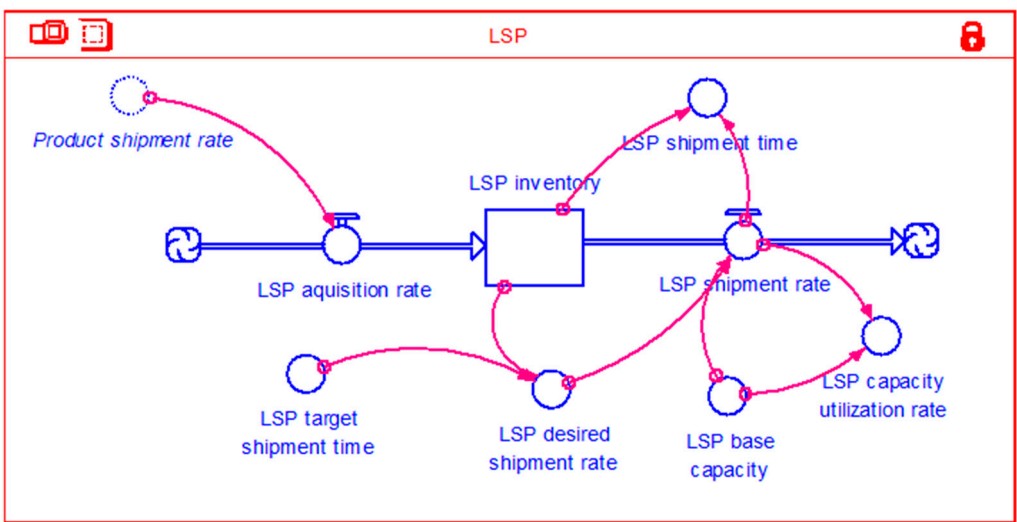

**Figure 3.** LSP's operation simulation model for Scenario 3.

Figure 4 shows the structure of the retailer's order fulfillment simulation model for Scenario 1. It consists of two material delays (for "Retailer inventory" and "Cumulative demand") and two information delays (for "Communicated lead time" and "Perceived delivery reliability"). "Customer demand rate" is a sum of "Real customer demand rate" and "Customer backlog adjustments". The retailer's managers will align the actual backlog ("Order backlog") and intended backlog ("Desired channel backlog"), over a certain time

("Time to adjust backlog"). "Desired channel backlog" is equal to "Real customer demand rate"×"LSP lead time expectation". The calculation of "LSP lead time expectation" is similar to that of "Product demand rate" in the producer's production simulation model, so the value of "Beta" is also set to be 0.5. "Inferred lead time" is equal to "LSP target lead time"×"Inferred capacity shortage". "Inferred capacity shortage" equals 1/"Perceived delivery reliability". "Current delivery reliability" is equal to "Normal delivery reliability" divided by the maximum value between 1 and "Workload". We also assume that the retailer can only collect the information of "Workload" one week after its managers perceive "LSP shipment time" from the LSP.

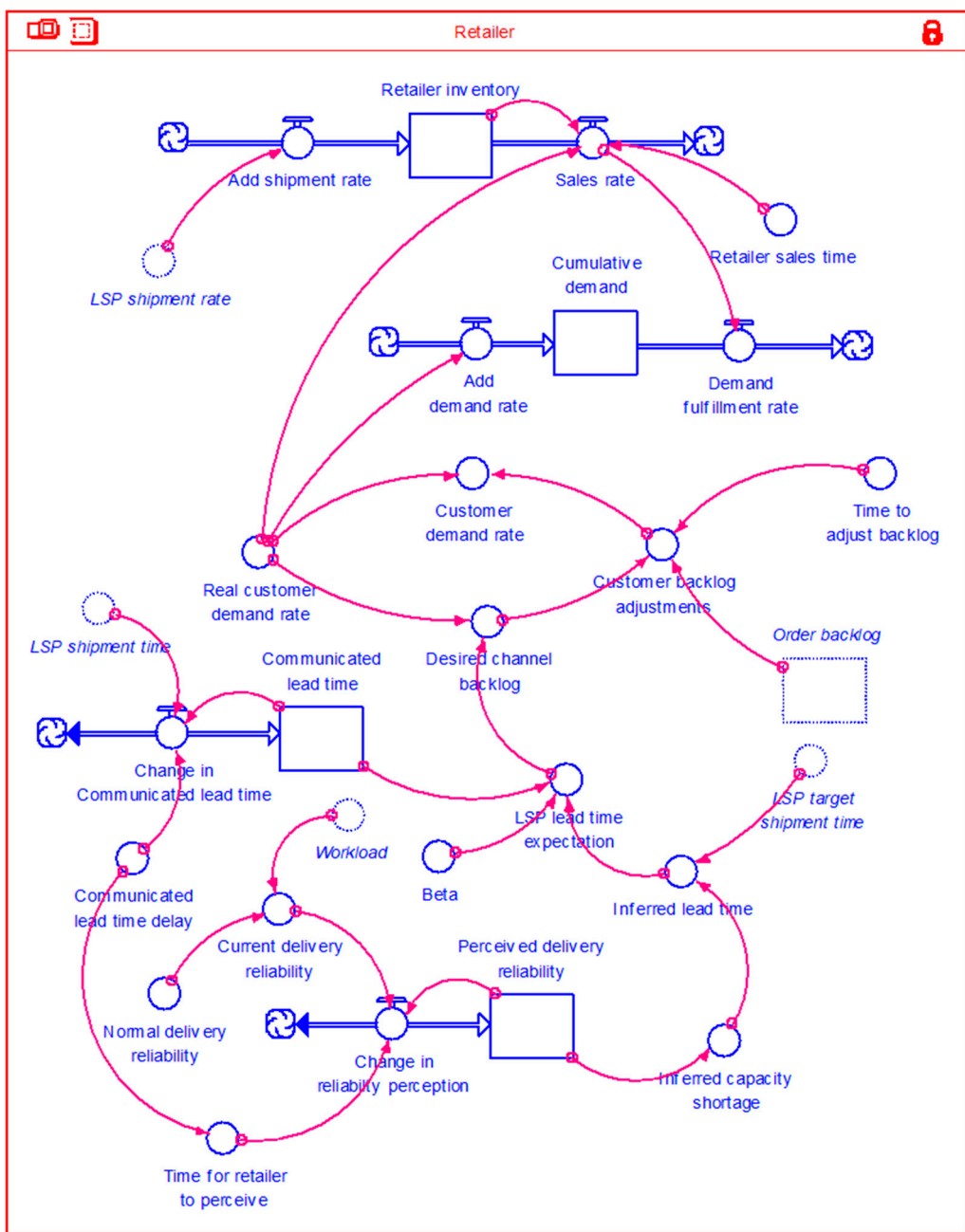

**Figure 4.** Retailer's order fulfillment simulation model for Scenario 1.

The retailer's order fulfillment simulation model for Scenario 2 is different from that for Scenario 1 on "LSP lead time expectation". Here, "LSP lead time expectation" is only from "Inferred lead time", which originates from an information delay of "Workload"

(the most proper information) that is directly shared by the LSP. In this sense, there is no "Communicated lead time" or "Beta". "Time for retailer to perceive" is replaced by "Communicated lead time delay" with the same expression and value in use as Scenario 1.

In the retailer's order fulfillment simulation model for Scenario 3, there are no processes for "Desired channel backlog" forecast or "Customer demand rate" generation, as the retailer just needs to provide "Real customer demand rate" to the producer. The other parts of the model are the same as those of Scenarios 1 and 2.

### 3.4. Analysis Methodology

We use a multiple analysis of variance (MANOVA) to compare the average of performance measures across structure scenarios for different SCI strategies after different disruptions. A Duncan test is used to identify significant differences in the mean values of a performance measure [58]. For the tests of parameter scenarios, only percentages of increase/decrease are reported, as the main purpose is to identify the trends to provide suggestions, rather than find the significant differences between parameter scenarios. In this study, we select two performance measures as the dependent variables for each firm: one focuses on supply efficiency, the other focuses on order fulfillment. From the supply chain's perspective, they reflect either the trade-off between capacity utilization and lead time [49] or the trade-off between inventory and backlog [42]. The selected six performance measures are: "Producer capacity utilization rate", "LSP capacity utilization rate", and "Retailer inventory" for supply efficiency; "Producer order backlog", "LSP shipment time", and "Cumulative demand" for order fulfillment. To evaluate the results of a producer (or an LSP) capacity disruption, better performance is indicated by higher values of "Producer capacity utilization rate" and "LSP capacity utilization rate". This is because higher capacity utilization rate justifies the investments in capacity. In contrast, "Producer capacity utilization rate" and "LSP capacity utilization rate" holds that the closer to 0.91 (which is the average value of "Real customer demand rate"/"Producer (or LSP) base capacity"), the better the performance. This is because extra capacity will not fulfill real customer demand, but generate extra costs. For the remaining four performance measures, it holds that lower values indicate better performance, when evaluating the results of all three types of disruptions. This is because the lower value leads to either lower costs or higher order fulfillment rates. The relative-precision procedure is used for sample size determination [59]. Ten runs for each scenario at 5% relative-precision level are suitable.

## 4. Results

### 4.1. The MANOVA Results

First of all, the Box's test results for all three types of disruptions (Table 3) are significant ($p < 0.001$) and indicate that homogeneity of variance-covariance is violated. So, Pillai's Trace statistic is used in interpreting the MANOVA results [60]. All three Pillai's Traces are significant ($p < 0.001$) and shown in Tables 4–6 for each type of disruptions.

**Table 3.** Box's test of equality of covariance matrices

|  | **Producer Capacity Disruption** | **LSP Capacity Disruption** | **Demand Disruption** |
|---|---|---|---|
| Box's M | 195 | 303 | 268 |
| F | 5.84 | 4.81 | 8.02 |
| df1 | 21.0 | 42.0 | 21.0 |
| df2 | $1.19 \times 10^3$ | $2.16 \times 10^3$ | $1.19 \times 10^3$ |
| *p*-value | 0.000 *** | 0.000 *** | 0.000 *** |

*** $p < 0.001$.

**Table 4.** The multiple analysis of variance (MANOVA) results for a producer capacity disruption

| Performance Measures | Mean (Standard Deviation) | | | F | *p*-Value | Duncan |
|---|---|---|---|---|---|---|
| | Scenario 1 | Scenario 2 | Scenario 3 | | | |
| Producer capacity utilization rate | 0.88 (0.04) | 0.87 (0.05) | 0.87 (0.05) | 0.35 | 0.706 | (1 2 3) |
| Producer order backlog | $8.22 \times 10^6$ ($5.69 \times 10^5$) | $8.00 \times 10^6$ ($6.86 \times 10^5$) | $8.00 \times 10^6$ ($6.86 \times 10^5$) | 0.36 | 0.699 | (1 2 3) |
| LSP capacity utilization rate | 0.80 (0.04) | 0.78 (0.05) | 0.79 (0.05) | 0.64 | 0.533 | (1 2 3) |
| LSP shipment time | 1.16 (0.09) | 1.13 (0.12) | 1.00 (0.00) | 7.03 | 0.003 ** | (1 2, 3) |
| Retailer inventory | $1.64 \times 10^6$ ($3.96 \times 10^5$) | $1.45 \times 10^6$ ($2.11 \times 10^5$) | $1.59 \times 10^6$ ($3.22 \times 10^5$) | 1.01 | 0.379 | (1 2 3) |
| Cumulative demand | $6.77 \times 10^5$ ($8.67 \times 10^5$) | $1.10 \times 10^6$ ($9.77 \times 10^5$) | $1.10 \times 10^6$ ($9.77 \times 10^5$) | 0.68 | 0.514 | (1 2 3) |

Overall test Pillai's Trace = 1.83 (F = 42.1, *p* < 0.001). ** *p* < 0.01.

**Table 5.** The MANOVA results for an LSP capacity disruption.

| Performance Measures | Mean (Standard Deviation) | | | F | *p*-Value | Duncan |
|---|---|---|---|---|---|---|
| | Scenario 1 | Scenario 2 | Scenario 3 | | | |
| Producer capacity utilization rate | 0.99 (0.03) | 0.97 (0.05) | 0.97 (0.05) | 0.50 | 0.610 | (1 2 3) |
| Producer order backlog | $8.37 \times 10^6$ ($8.36 \times 10^5$) | $7.89 \times 10^6$ ($9.13 \times 10^5$) | $7.89 \times 10^6$ ($9.13 \times 10^5$) | 0.97 | 0.392 | (1 2 3) |
| LSP capacity utilization rate | 0.87 (0.02) | 0.85 (0.03) | 0.85 (0.04) | 0.48 | 0.627 | (1 2 3) |
| LSP shipment time | 1.25 (0.08) | 1.20 (0.05) | 1.17 (0.05) | 4.54 | 0.020 * | (1 2, 2 3) |
| Retailer inventory | $1.73 \times 10^6$ ($4.29 \times 10^5$) | $1.60 \times 10^6$ ($2.46 \times 10^5$) | $1.63 \times 10^6$ ($3.38 \times 10^5$) | 0.38 | 0.686 | (1 2 3) |
| Cumulative demand | $1.24 \times 10^6$ ($1.22 \times 10^6$) | $1.25 \times 10^6$ ($1.21 \times 10^6$) | $1.24 \times 10^6$ ($1.22 \times 10^6$) | 0.00 | 1.000 | (1 2 3) |

Overall test Pillai's Trace = 1.01 (F = 3.89, *p* < 0.001). * *p* < 0.05.

**Table 6.** The MANOVA results for a demand disruption.

| Performance Measures | Mean (Standard Deviation) | | | F | *p*-Value | Duncan |
|---|---|---|---|---|---|---|
| | Scenario 1 | Scenario 2 | Scenario 3 | | | |
| Producer capacity utilization rate | 0.96 (0.04) | 0.91 (0.06) | 0.91 (0.06) | 2.38 | 0.112 | (1 2 3) |
| Producer order backlog | $7.74 \times 10^6$ ($6.75 \times 10^5$) | $7.12 \times 10^6$ ($6.72 \times 10^5$) | $7.06 \times 10^6$ ($6.32 \times 10^5$) | 3.23 | 0.055 | (1, 2 3) |
| LSP capacity utilization rate | 0.93 (0.03) | 0.91 (0.05) | 0.91 (0.06) | 0.83 | 0.447 | (1 2 3) |
| LSP shipment time | 1.85 (0.17) | 1.78 (0.29) | 1.00 (0.00) | 59.8 | 0.000 *** | (1 2, 3) |
| Retailer inventory | $2.07 \times 10^6$ ($6.77 \times 10^5$) | $1.80 \times 10^6$ ($3.87 \times 10^5$) | $2.26 \times 10^6$ ($4.13 \times 10^5$) | 2.10 | 0.141 | (1 2 3) |
| Cumulative demand | $4.29 \times 10^5$ ($8.08 \times 10^5$) | $4.34 \times 10^5$ ($8.18 \times 10^5$) | $2.98 \times 10^5$ ($5.20 \times 10^5$) | 0.11 | 0.895 | (1 2 3) |

Overall test Pillai's Trace = 1.90 (F = 69.6, *p* < 0.001). *** *p* < 0.001.

For the first two types of disruptions, we will focus on order fulfillment measures ("Producer order backlog", "LSP shipment time", and "Cumulative demand"), as the main problem of these disruptions is the lack of capacity, leading to delayed orders (measured by "Producer order backlog", "LSP shipment time", and "Cumulative demand") and/or low-quality products (shown by "LSP shipment time", as cheese is a perishable

product). In contrast, supply efficiency measures ("Producer capacity utilization rate", "LSP capacity utilization rate", and "Retailer inventory") are the main concerns for the demand disruption.

The MANOVA results for a producer capacity disruption are shown in Table 4. Only "LSP shipment time" is found to be significantly different among three structure scenarios ($p < 0.01$). Through a Duncan test, we group similar structure scenarios and arrange the groups from low to high performance [61]. The same logic applies to the Duncan test results for an LSP capacity disruption in Table 5 and a demand disruption in Table 6. Here, the Duncan test result of "LSP shipment time" shows that Scenario 3 is the best SCI strategy to deal with a producer capacity disruption. There is no significant difference between Scenarios 1 and 2.

Table 5 illustrates the MANOVA results for an LSP capacity disruption. Only "LSP shipment time" is found to be significantly different among three structure scenarios with $p < 0.05$. This is also verified by the Duncan test results, as the only significant difference is found in the result of "LSP shipment time". It shows that Scenario 3 is the best SCI strategy to deal with an LSP capacity disruption, while Scenario 1 is the worst SCI strategy. There is no significant difference between Scenarios 1 and 2 or between Scenarios 2 and 3.

The MANOVA results for a demand disruption are presented in Table 6. Only "LSP shipment time" is found to be significantly different among three structure scenarios with $p < 0.001$. There are two Duncan test results that demonstrate separated groups. The Duncan test result of "LSP shipment time" illustrates that Scenario 3 is better than Scenarios 1 and 2. Although "Producer order backlog" is not found to be significantly different among three scenarios, its Duncan test result shows that Scenarios 2 and 3 are better than Scenario 1. All in all, Scenario 3 is the best SCI strategy to deal with a demand disruption, while Scenario 2 ranks second and Scenario 1 ranks last.

*4.2. The Parameter Scenario Testing Results*

The parameter scenario testing results for "Alpha" (from zero to one) of Scenario 1 show that, after a producer capacity disruption, "Cumulative demand" can be fully eliminated (100%) with some expenses (<25%) of the other five performance measures. As "Cumulative demand" belongs to order fulfillment measures, increasing "Alpha" may be a more beneficial choice for recovery. For an LSP capacity disruption, only "Producer order backlog" increases 18%, thus decreasing "Alpha" is more helpful for supply chains of Scenario 1. Except for "Cumulative demand", all the other five performance measures have increases in the context of a demand disruption recovery. Therefore, decreasing "Alpha" is more favorable for supply chains of Scenario 1 to recover from a demand disruption.

The parameter scenario testing results for "Beta" (from zero to one) of Scenario 1 demonstrate no value change in an LSP capacity disruption, thus no change has to be made on "Beta" for this type of disruption. For a producer capacity disruption, 4% decrease in "Retailer inventory" is found at the expense of 4% increases in both "LSP shipment time" and "Cumulative demand". As "LSP shipment time" and "Cumulative demand" are order fulfillment measures, it is better to decrease "Beta". For a demand disruption, "LSP shipment time" drops 14%, with the increases in the other four performance measures, especially 21% in "Retailer inventory". As we pay more attention to supply efficiency measures ("Producer capacity utilization rate", "LSP capacity utilization rate", and "Retailer inventory") for a demand disruption, decreasing "Beta" is recommended.

The parameter scenario testing results for "Producer perception delay" (from 1 week to 3 weeks) are available for both Scenarios 1 and 2. For Scenario 1, "Cumulative demand" decreases sharply (66%) after a producer capacity disruption, with slight improvements (<10%) in the other performance measures. As "Cumulative demand" belongs to order fulfillment measures, prolonging "Producer perception delay" is more beneficial to supply chains of Scenario 1 to recover from a producer capacity disruption. For an LSP capacity disruption, only "Producer order backlog" increases 7%, thus shortening "Producer perception delay" is more helpful. For a demand disruption, "Producer capacity utilization

rate" and "Producer order backlog" increase slightly (<10%), while "LSP shipment time" increases 15%. Shortening "Producer perception delay" is more favorable for supply chains of Scenario 1 to recover from a demand disruption. For Scenario 2, only "LSP shipment time" increases 2% in the context of a producer capacity disruption recovery, therefore shortening "Producer perception delay" is more helpful. No change is found in an LSP capacity disruption, thus no change should be made on "Producer perception delay". For a demand disruption, "LSP capacity utilization rate" and "Retailer inventory" drop 1% and 7%, respectively, with the increase in "LSP shipment time" of 8%. As we pay more attention to supply efficiency measures ("Producer capacity utilization rate", "LSP capacity utilization rate", and "Retailer inventory") for a demand disruption, prolonging "Producer perception delay" is recommended.

According to the parameter scenario testing results for "Communicated lead time delay" (from 1 week to 3 weeks), changes are only found in a demand disruption for both Scenarios 1 and 2, thus no change has to be made on "Communicated lead time delay" for the other two types of disruptions. For Scenario 1, "Producer capacity utilization rate", "Producer order backlog", and "LSP capacity utilization rate" all increase 1%, with a rise in "LSP shipment time" of 9%. Therefore, shortening "Communicated lead time delay" is more favorable for supply chains of Scenario 1 to recover from a demand disruption. For Scenario 2, "Retailer inventory" drops 7% at the expense of 7% increase in "LSP shipment time". As "Retailer inventory" is a supply efficiency measure, prolonging "Communicated lead time delay" is more beneficial to supply chains of Scenario 2 to recover from a demand disruption.

## 5. Discussions and Conclusion

### 5.1. Discussions

This paper distinguishes three SCI strategies with different dimensional focuses: information integration (Scenario 1), relational integration (Scenario 2), and operational integration (Scenario 3). Our results indicate that Scenario 3 is the best SCI practice, regardless of any type of disruption, while Scenario 1 usually achieves the worst performance. The significant differences are from "LSP shipment time" (for all types of disruptions: Tables 4–6) and "Producer order backlog" (for a demand disruption: Table 6), which are in line with Van der Vorst et al.'s [62] findings that long lead time causes more order backlog and both of them have significant impacts on perishable food supply chain performance. Our findings are also consistent with an evolutionary perspective of SCI: information integration gives firms a competitive advantage as the first step. Working as partners to share the most appropriate information leads to greater benefits [16]. We extend this perspective that further elimination of information delay (i.e., full information transparency) helps the supply chain achieve the best performance.

The differences between Scenarios 1 and 2 result from the accuracy and integrity of information exchanged amongst the firms [14,15]. Compared to Scenario 2 of sharing "Real customer demand rate" and "Workload", Scenario 1 mainly shares "Channel demand rate" and "LSP shipment time", although information of "Real customer demand rate" and "Workload" can be collected through other ways for a longer time. The information sharing of "LSP shipment time" to the retailer leads to stronger fluctuation in "LSP lead time expectation" (Figure 4) that in turn leads to stronger oscillation of orders that serve as the pipeline inventory [43]. Finally, the pipeline inventory adds to "Producer order backlog" through sharing "Channel demand rate" (Figure 1). This dynamic mechanism is supported by the parameter scenario testing results of "Alpha" and "Beta". As increasing "Alpha" or "Beta" leads to more "Channel demand rate" or "LSP shipment time" that is shared, "Producer order backlog" always surges as a result. Moreover, compared to the parameter scenario testing results of "Producer perception delay" and "Communicated lead time delay", those of "Alpha" and "Beta" surge more. Such findings highlight that timely information exchange alone is not enough [63]. To leverage the business value of

information integration, relational mechanisms should be devised to enhance the sharing of accurate and reliable information along the supply chain [64].

Scenario 3 generates the lowest "Producer order backlog" and "LSP shipment time" in all types of disruptions (Tables 4–6), as, compared to Scenario 2, Scenario 3 further eliminates information delay. This is supported by the parameter scenario testing results of "Producer perception delay" and "Communicated lead time delay". When "Producer perception delay" and "Communicated lead time delay" are shortened, "Producer order backlog" and "LSP shipment time" always drop as a result. Since facilitating full information transparency reduce order oscillation that is caused by information distortion and information delay [52], operational integration (Scenario 3) functions the best in dealing with supply chain disruptions.

There are two situations that seem counterintuitive. First, when increasing "Alpha" or "Producer perception delay", "Cumulative demand" can be significantly decreased with moderate or slight increases on the other five performance measures in a producer capacity disruption of Scenario 1. Therefore, increasing "Alpha" and "Producer perception delay" may be more beneficial to the supply chain. This situation is very special, as both the producer and the LSP have a surge capacity. "Producer surge capacity" can serve as a first aid to a producer capacity disruption [21], while "LSP surge capacity" helps speed up goods-in-transit [65]. By increasing "Alpha" or "Producer perception delay", the producer will receive less order adjustments that are based on the retailer's perception of the producer's capacity shortage. Both "Producer surge capacity" and "LSP surge capacity" can thus be fully used to fulfill customer demand, ending up with much less "Cumulative demand". As a result, the supply chain can achieve a competitive advantage on current order fulfillment rate (based on less "Cumulative demand"), but not on future order fulfillment rate (based on less "Producer order backlog") or on product quality (based on shorter "LSP shipment time").

The second remarkable situation becomes apparent from the parameter scenario testing results of "Producer perception delay" and "Communicated lead time delay" in a demand disruption of Scenario 2. "Retailer inventory" drops at the expense of an increase in "LSP shipment time". As we pay more attention to supply efficiency measures ("Producer capacity utilization rate", "LSP capacity utilization rate", and "Retailer inventory") for a demand disruption recovery, prolonging "Producer perception delay" and "Communicated lead time delay" are recommended. The supply chain can thus achieve a competitive advantage on saving costs (based on less "Retailer inventory"), but not on order fulfillment and product quality (based on shorter "LSP shipment time").

*5.2. Managerial Implications*

First and foremost, from the performance angle, firms and supply chains are highly encouraged to follow the evolution of SCI strategies (from Scenario 1 to Scenario 3) to better cope with supply chain disruptions in a long run. Furthermore, at the short term, we recommend the following. For Scenario 1, both collecting accurate and reliable information and shortening information delay are more favorable to both individual firms and the supply chain as a whole. Therefore, firms are suggested to follow their own interests to improve certain factors. In detail, the producer should develop more ways to collect "Real customer demand rate" as quickly as possible; the LSP should try its best to shorten "Producer perception delay" and "Communicated lead time delay"; the retailer should find more sources to quickly collect the information of "Workload". However, a producer capacity disruption is a special case, thus facilitating a different solution. If the supply chain wants to excel in a high level of customer satisfaction after a producer capacity disruption, blurring the transparency of "Real customer demand rate" to the producer is recommended. As this solution incurs more costs for all firms, they should agree on the same purpose of keeping a high level of customer satisfaction. For Scenario 2, shortening information delay is more favorable to both the LSP itself and the supply chain as a whole. Thus, the LSP should try its best to shorten "Producer perception delay" and "Communicated

lead time delay", while the producer and the retailer are encouraged to provide help. However, prolonging information delay is more beneficial to the whole supply chain in a demand disruption. Although at the expense of longer "LSP shipment time", "Retailer inventory" can become lower and thus saving costs, which is the core issue when demand is weak. To apply this solution successfully, the retailer should share some of its cost savings to the LSP to motivate information delay, as it benefits most from this solution. Last but not least, Scenario 3 performs best. It also requires efforts from all firms. They should jointly re-define the terms of their relationships through practices such as risk pooling, revenue sharing, and balanced scorecard.

*5.3. Conclusion, Limitations, and Future Research*

Since the introduction of supply chain management, one of the main themes has been the crucial role of SCI in performance improvement [66,67]. Nowadays, global competition has shaped complex and tightly coupled inter-firm networks in which disruptions in the flows of materials, information, and funds have become the norm [68]. This requires closer integration to ensure that the flows of product, information, and payments operate efficiently [9,18,32]. Although a growing body of research in operations management, marketing, and accounting analyzes the benefits of SCI, prior research does not offer a comprehensive examination of the effects of different SCI strategies on disruption recovery. Through a system dynamics simulation, our paper is among the first to present and compare the dynamic mechanisms of different SCI strategies, support and extend the evolutionary perspective of SCI, and provide suggestions for supply chains using different SCI strategies.

Every study has limitations which give rise to avenues for further research. Our paper is based on a group of simulation models with specific problem assumptions and parameter settings, which imposes boundaries and limitations to our analyses. First, although our findings clearly show an evolutionary perspective of SCI, we did not model the way of evolution between different SCI strategies. Akkermans et al. [63], for example, have addressed the question of how to move from limited information sharing to full transparency. The Cyber-Physical Systems (CPS) approach would also provide a framework to understand the way of evolution, as well as the crucial role of human resource management [69]. Whether and how such research can be aligned with our models will be worth researching in the future. Second, although cost issues were not the main focus of this paper, they are greatly related to firms' decision-making [70]. In order to help firms share risks, costs, and rewards equitably, carefully designed cost variables are highly recommended to include in future models.

**Author Contributions:** Conceptualization, Q.Z.; methodology, Q.Z.; software, Q.Z.; validation, Q.Z.; formal analysis, Q.Z.; investigation, Q.Z.; resources, H.K.; data curation, Q.Z. and H.K.; writing—original draft preparation, Q.Z.; writing—review and editing, Q.Z., H.K. and M.C.J.C.; visualization, Q.Z.; supervision, H.K. and M.C.J.C.; project administration, Q.Z. All authors have read and agreed to the published version of the manuscript.

**Funding:** This research received no external funding.

**Institutional Review Board Statement:** Not applicable.

**Informed Consent Statement:** Not applicable.

**Data Availability Statement:** Not applicable.

**Conflicts of Interest:** The authors declare no conflict of interest.

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
