# Peer review of "The Effects of Different Supply Chain Integration Strategies on Disruption Recovery: A System Dynamics Study on the Cheese Industry"

_logistics, 2021_

Round 1

Reviewer 1 Report

The work has an important simulation.

The authors present relevant results.

The article has an interesting structure.

I indicate to the authors the following recommendations:

Authors must indicate the period of time to collect the analyzed information.

The bibliographic review is interesting. However, authors must include the most recent bibliographic references.

Authors should include study limitations in conclusions

Authors must adapt to the journal format. Authors should review citations and bibliographic references.

Author Response

Point 1: Authors must indicate the period of time to collect the analyzed information.

Response 1: Thank you for this suggestion! Our data were collected in 2011. Cost structures have not been changed significantly since then. Such information has been added to lines 146-148 in our revised manuscript.

Point 2: The bibliographic review is interesting. However, authors must include the most recent bibliographic references.

Response 2: Thank you for this comment! Indeed, we need to include more recent references, which have been added to the revised manuscript. The newly added references are (to keep the listing simple, we only report each reference number here): 27, 28, 37, 46, 60, 69, and 70.

Point 3: Authors should include study limitations in conclusions.

Response 3: Thank you for your suggestion! In the revised manuscript, we have separated "limitations and future research" part as a new paragraph, and enriched with new literature (reference number: 69 and 70).

Point 4: Authors must adapt to the journal format. Authors should review citations and bibliographic references.

Response 4: Thanks for your kind reminder! We have carefully adjusted the format to journal requirements in our revised manuscript.

Reviewer 2 Report

Dear authors, 

I really thank you for the opportunity to read your paper "The effects of different supply chain integration strategies on disruption recovery: a system dynamics study on the cheese industry". I hope you and your family are well in this particular pandemic period. 

Looking at your paper : 

a) I like your paper, but I think that a little consideration about the use of the new industry in the logistic can be done (also from a management point of view). In this regard, I suggest these authors (I selected for all the review the same, so you have not to search a lot) :

1) https://doi.org/10.3390/su12104075 

2) https://doi.org/10.7232/iems.2020.19.3.551 (in this paper, you have a new logic that I think you can also use for future research). 

b) About System Dynamics: I really like the use of SD for management. I'd like to start to work on me too. Meanwhile, I suggest these papers (about the use of management in different fields): 

3) https://doi.org/10.3390/app10030903 (safety Management system - thanks to System Dynamics). 

4)https://doi.org/10.15866/iremos.v10i1.11133 (always on SD). 

I think that 1,3,4 has to be included in the literature and 2 can help you for future works. 

Please cite all these papers to improve your research and reinforce it. 

In particular, I find the second paper suggested interesting for future research even if the particular case study is classic but analyzed with innovation. So I think the future has to consider costs as you correctly state in the conclusions (3rd paper suggested also consider them) and a new challenge. In this regard, I find the future research quite weak (also in the literature you consider. This considered literature is important but a little dated, don't you?). 

C) look at the formatting of your tables/figures and English revisions.

I think that you will get easily the result!

Author Response

Point 1: I like your paper, but I think that a little consideration about the use of the new industry in the logistic can be done (also from a management point of view). In this regard, I suggest these authors (I selected for all the review the same, so you have not to search a lot):

1) https://doi.org/10.3390/su12104075

2) https://doi.org/10.7232/iems.2020.19.3.551 (in this paper, you have a new logic that I think you can also use for future research).

Response 1: Many thanks! We have cited them in the revised manuscript at lines 91-94 (the first paper, which enriches our literature review) and lines 563-565 (the second paper, which is for sure a thoughtful framework for future research).

Point 2: About System Dynamics: I really like the use of SD for management. I'd like to start to work on me too. Meanwhile, I suggest these papers (about the use of management in different fields):

3) https://doi.org/10.3390/app10030903 (safety Management system - thanks to System Dynamics).

4)https://doi.org/10.15866/iremos.v10i1.11133 (always on SD).

I think that 1,3,4 has to be included in the literature and 2 can help you for future works.

Please cite all these papers to improve your research and reinforce it.

In particular, I find the second paper suggested interesting for future research even if the particular case study is classic but analyzed with innovation. So I think the future has to consider costs as you correctly state in the conclusions (3rd paper suggested also consider them) and a new challenge. In this regard, I find the future research quite weak (also in the literature you consider. This considered literature is important but a little dated, don't you?).

Response 2: Again, many thanks for your suggestions! We have cited the third paper at lines 566-567, which gives us a solid and recent reference for our argument. The fourth paper is cited at lines 115-118, which helps us enrich our system dynamics literature.

Point 3: look at the formatting of your tables/figures and English revisions.

Response 3: Thanks for your kind reminder! We have carefully adjusted the format to journal requirements in our revised manuscript.

Round 2

Reviewer 2 Report

Congratulations